# Oral and Dental Considerations of Combat-Induced Post Traumatic Stress Disorder (PTSD)—A Cross-Sectional Study

**DOI:** 10.3390/jcm11113249

**Published:** 2022-06-06

**Authors:** Nirit Tagger-Green, Carlos Nemcovsky, Naama Fridenberg, Orr Green, Liat Chaushu, Roni Kolerman

**Affiliations:** 1Department of Periodontology and Oral Implantology, The Maurice and Gabriela Goldschleger School of Dental Medicine, Tel Aviv University, Tel Aviv 6997801, Israel; carlos@tauex.tau.ac.il (C.N.); liatnata@post.tau.ac.il (L.C.); ronik@tauex.tau.ac.il (R.K.); 2Department of Oral Rehabilitation, The Maurice and Gabriela Goldschleger School of Dental Medicine, Tel Aviv University, Tel Aviv 6997801, Israel; namifrid@gmail.com; 3Sackler School of Medicine, Tel Aviv University, Tel Aviv 6997801, Israel; orrgreen@gmail.com

**Keywords:** combat PTSD, stress disorders, oral medicine, periodontal disease, oral hygiene, tooth wear

## Abstract

Objective: This study compared dental, periodontal, oral, and joint/muscle tenderness among Israeli combat-induced post-traumatic stress disorder (Ci-PTSD) war veterans to non-PTSD patients. Study design: This retrospective three-arm study compared oral and facial manifestations between 100 Israeli veterans with Ci- PTSD (study group) and 103 non-PTSD periodontal patients (Control group). The study group was further divided into two subgroups of individuals who received psychiatric medications (40 patients) or did not (60 patients). All patients underwent complete dental, oral, and periodontal examinations, including assessing signs of parafunction. Results: All PTSD patients had poor oral hygiene. The plaque index (PI) was higher in the PTSD group compared to the control group (0.72 ± 0.28 vs. 0.45 ± 0.29, respectively, *p* < 0.001). The decayed, missing, and filled teeth score (DMFt) was higher in the PTSD population than in the controls (19.97 ± 8.07 vs. 13.05 ± 6.23 *p* < 0.05). Severe periodontal disease was more common among the PTSD subgroup taking medications (med -group) (62.5%) compared to the nonmedicated group (non-med group) (30.0%) and the controls (27.2%) (*p* = 0.001). Heavy smoking was more prevalent in the medicated PTSD patients than in other groups. Conclusions: The present study shows higher morbidities in combat-induced PTSD patients, including oral, dental, and periodontal manifestations, especially in medicated patients.

## 1. Introduction

Post-traumatic stress disorder (PTSD) is one of the most common psychiatric illnesses in the United States. PTSD is a mental health condition triggered by a terrifying event. Symptoms may include flashbacks, nightmares, severe anxiety, and uncontrollable thoughts about the event [1,2]. The reported prevalence of PTSD is 3.6–19% [1,2,3,4]. The prevalence of combat-induced PTSD (Ci-PTSD) in Israel ranges between 1.5% [5] and 6.85% [6].

PTSD may be associated with various medical conditions, including sleep and respiratory disorders, osteoporosis, migraine, and diabetes [7,8]. Facial and temporomandibular pain and headache may be detected as early signs in 88% of PTSD patients [9]. Abfraction, abrasion, and attrition of enamel and dentin loss are frequent findings among these patients [10,11]. Our previous pilot study reported a high incidence of poor oral hygiene, rampant caries, multiple missing teeth, severe periodontal disease, and temporomandibular disorders (TMDs) [10]. Xerostomia, changes in taste, glossitis, gingivitis, and periodontitis have also been found in 15–25% of PTSD patients [10,11,12].

Chronic periodontitis is a highly prevalent disease of tooth-supporting tissues, leading to alveolar bone loss and, finally, tooth loss if left untreated. It is also associated with low-grade systemic inflammation [13] that is either directly induced via dislocation of periopathogenic bacteria into the bloodstream or indirectly via increased levels of locally produced proinflammatory mediators [14]. Periodontal treatment is based, first and utmost, on motivation to undertake successful removal of supragingival dental biofilm by the patient and all measures headed toward risk factor control (smoking, glycaemic control, etc.) (step-1) and complete removal of subgingival biofilms and associated calculus deposits by scaling and root planning (SRP) (step-2). The response to these two steps commonly referred to as cause-related therapy, should be assessed once the periodontal tissues have healed (periodontal re-evaluation), usually between 6 and 12 weeks after completing the first two steps. Periodontal surgery that must be considered as an adjunctive to cause-related therapy is aimed at treating those areas of the dentition not responding adequately to the first two steps (presence of pockets >4 mm with BOP (bleeding on probing) or presence of deep periodontal pockets (≥6 mm). The primary purpose of this therapy step is to gain further access to subgingival instrumentation or in those lesions that add complexity to the management of periodontitis (intrabony and furcation lesions) to either regenerate or resect them [15].

Due to the well-established association between periodontitis and other systemic diseases [16,17,18,19], these findings may have significant medical implications. Yet, dental and facial morbidities are also common in the general population.

The present study aimed to compare the dental, periodontal, oral morbidities, and joint/muscle tenderness of Israeli veterans diagnosed with Ci-PTSD to an age- and gender-matched control group non-PTSD periodontal patients. Furthermore, within the PTSD group, the association between the use of psychiatric medications and the oral manifestations was also evaluated.

## 2. Materials and Methods

### 2.1. Study Population

The study included 203 individuals (199 males and four females) divided into three groups. The study group consisted of 100 consecutive patients (97 males and three females) referred during 2015–2019 for periodontal or implant treatment to the senior author (NTG). All patients were diagnosed at least ten years earlier and met the criteria for severe PTSD according to the 4th edition of the Diagnostic and Statistical Manual of Mental Disorders (DSM-4). This group was further divided into two subgroups: 60 patients who were not treated with psychiatric medications (58 males and two females)—in the non-med group and 40 patients treated with psychiatric drugs (39 males and one female) —in the med group. (All the patients in the med-group would not have been able to function without psychiatric medication).

The control group comprised 103 consecutive patients (102 males and one woman) referred for periodontal or implant treatment to the same periodontist (NTG) in 2019. None of these patients reported the use of psychiatric medications.

All patients in the study and control groups agreed to participate and signed informed consent. The study was approved by the Institutional Review Board (IRB) of Tel-Aviv University.

All participants completed a self-administered medical and sociodemographic written closed (yes/no) questionnaire concerning joint/muscle tenderness and/or pain, the use of a nightguard, past medical history, medications, allergies, and smoking habits. After the patient filled out the questionnaire, the same periodontist went over the questions to clarify essential remarks. In addition, patients were classified according to their smoking status as never smoked, past smokers, current light smokers (≤10 cigarettes a day), and current heavy smokers (>10 cigarettes a day).

### 2.2. Patients Examination

All patients underwent a thorough evaluation, including a full-mouth periodontal chart, occlusal analysis, and radiological examinations employing panoramic radiograph and/or a complete set of periapical radiographs. Additionally, all the patients underwent a thorough extra and intraoral examination performed by the senior author (NTG). Since it was a cross-sectional study, all clinical outcomes were recorded upon acceptance to the office- during the first appointment.

### 2.3. Extraoral Examination

The patients were examined for functional and nonfunctional joint/muscle tenderness. Patients were considered to have a positive finding if at least one of the following was detected:(a)The pain was localized around the temporomandibular joint (TMJ), temporalis, or masseter muscles.(b)The patient reported pain in the TMJ, aggravated by palpation of the lateral pole or around it when the mouth was closed.(c)Aggravation of the pain during maximum assisted or unassisted opening or lateral/protrusive movements [20,21].

### 2.4. Intraoral Examination

The following variables were assessed:Probing depth (PD), measured using a 15-UNC probe and recorded to the nearest 1 mm (PCP- UNC 15; Hu- Friedy, Chicago, IL, USA) at 6 points around all teeth.Plaque index (PI) assessed the amount of visible dental plaque [22].The DMFt (decayed-D, missing-M, and filled-F teeth) score (ranging from 0–28) assesses the prevalence and treatment needs of dental caries [23]. The score is the sum of carious, absent, and obturated teeth. The third molars were not considered in the count.Intraoral hard and soft tissue pathologies.Occlusal tooth wear is classified into three categories: 0: no wear; 1: wear confined to the enamel; and 2: severe—wear with exposed dentin [24].Number of dental implants.

### 2.5. Periodontal Diagnosis

The patients were diagnosed with periodontitis if they met the European Federation of Periodontology criteria and the American Academy of Periodontology in 2018 [25]. Diagnosis of localized periodontitis was established if <30% of the sites were involved, whereas generalized periodontitis was diagnosed if ≥30% of the sites were affected. The four stages were determined considering clinical attachment loss, amount and percentage of bone loss, PD, presence, the extent of angular bony defects, furcation involvement, tooth mobility, and tooth loss due to periodontitis.

### 2.6. Statistical Analysis

The statistical analyses were performed using SPSS 20.0 statistical analysis software (SPSS Inc., Chicago, IL, USA). The measurements of the categorical variables are presented as counts (i.e., absolute frequencies) in each of the categories and as percentages of all the counts in the whole relevant set of categories (i.e., relative frequencies). The continuous variables are presented by their means and standard deviations.

In addition, comparisons (planned and post-hoc) were performed between the whole PTSD group and the control group and between the two subgroups within the PTSD population and the control group. Categorical variables were analyzed between groups using the Pearson Chi-Square test. Continuous variables were compared using the *t*-test (for PTSD vs. controls) and ANOVA test (for the three patients groups: PTSD patients with/without medications and controls). For statistically significant differences, posthoc analysis was performed by TukeyB test.

When the measures of quantitative variables were not distributed normally, we used non-parametric tests: the Wilcoxon-Mann-Whitney test (between the whole PTSD group and the Control group) and the Kruskal-Wallis test (between the three groups of patients. This was relevant only for the PI, as the remaining quantitative variables were normally distributed.

A two-sided *p* value of < 0.05 was considered statistically significant for all analyses.

## 3. Results

Table 1 presents the demographic variables and the prevalence of dental findings among PTSD patients with/without medications and a control group of periodontal patients without psychiatric disorders.

While statistically significant differences were found in PI and DMF between patients groups, the rate of dental implants was similar in the three groups of patients.

The average PI was the highest among the PTSD med. group (0.80), followed by the non-med group (0.67) and the control group (0.45). All the differences between the groups were statistically significant using the ANOVA and *t*-tests.

However, PI values were not normally distributed (Figure 1). Most of the values were either very high (PI > 0.7) or extremely low (PI < 0.2). Therefore, as detailed above, we used non-parametric tests that confirmed the statistically significant differences between the groups (*p* value < 0.05).

DMFt scores of the study group (whole PTSD group) were significantly higher than in the control group.

The differences in the DMFt score were mainly attributed to the filled (F) teeth in all groups. The difference between the study and control groups was attributed to a relatively small average number of decayed teeth in the control group, 0.45, versus the average of 3.03 for the non-med subgroup and 4.13 for the med subgroup.

Table 2 summarizes the smoking status, oral, dental, and periodontal manifestations among the three groups of patients.

All systemic diseases were screened and recorded. The most common were ischemic heart disease (IHD) in 9.7% of the control group and 7.5% of the PTSD group and diabetes mellitus (DM) in 7.8% of the control group and 6.7% of the PTSD group. Both IHD and DM were found in 2.9% of the control group and 3.3% of the PTSD group. Statistical analysis was not performed due to the low prevalence of systemic diseases in all three patients’ groups.

The percentage of heavy smokers was higher in the med group (22 patients, 55.0%) than in the non-med group (13 patients, 21.7%) and the control group (16 patients, 15.5%). In addition, a positive association between heavy smoking and PTSD was found, especially in the med-group (*p* value < 0.001).

Severe periodontal disease was more prevalent in the med group (25 patients 62.5%) than in the non-med group (18 patients 30%) and control group (28 patients 27.2%). Statistically significant differences (*p* value < 0.001) were found in the severity of periodontal disease (two categories, mild and severe).

The intervening masking effect of smoking was removed to examine the possible uncorrelated influence of PTSD on the severity of the periodontal disease. We have used a Logistic regression of the odds ratio of having severe periodontal disease (as the dependent variable) on the heavy smoking and non-med and med groups (as the predictors). The Logistic regression demonstrated that the odds ratio of having severe periodontal disease among heavy smokers was 4.11 (95% CI = 1.373 to 7.774). The med- group’s severe periodontal disease odds ratio was 3.27 (95% CI = 1.990 to 8.500). Thus, heavy smoking elevates the incidence of generalized periodontal disease.

## 4. Discussion

In this study, we agree with previous reports, that most patients with Ci- PTSD have poor oral hygiene and periodontal disease [8,10]. Furthermore, the high prevalence of severe periodontitis in PTSD patients was attributed to the neglect of oral hygiene, smoking habits, and altered immune response [3,7].

We found that periodontitis was more severe among patients treated with psychiatric medication than non-medicated Ci-PTSD patients and the control group.

The rate of severe periodontal disease in the control group was slightly higher than the reported 11% global prevalence [26], but higher than in the USA (8.9%) [27]. In Germany, the report in 2014 was between 16.9–48%, depending on the definition of severe periodontal disease [28]. In a previous study [13], we found a significant correlation between smoking status and severe periodontal disease, in accordance with other publications [28,29,30]. However, the present study shows, for the first time, that despite the high percentage of smokers among PTSD patients, Ci-PTSD is an independent factor associated with severe periodontitis. This may be attributed to the altered immune response in PTSD patients that might affect the susceptibility to severe periodontal disease [2,4]. Most patients in the present cohort had symptoms of TMJ dysfunction and/or orofacial pain. The highest percentage of this feature was found in the med group, lower in the non-med group, and very low in the controls (*p* = 0.001). A high prevalence of facial pain has been previously reported in psychiatric patients, including PTSD [2,3,12,31,32]. Awake or sleep bruxism is considered a contributing factor to the worsening of periodontal disease [31,32]. Although it is controversial whether traumatic occlusal forces lead to periodontal attachment loss, its association with non-carious cervical lesions was suggested [33]. Abnormal attrition, muscle, and joint sensitivity are significantly more common in psychiatric patients than in controls (46.8% and 20%, respectively) [33,34].

The high percentage of reported facial pain in the med group compared to the non-med group may be related to the high rate of patients using selective serotonin (SSRIs) and norepinephrine (SNRIs) reuptake inhibitors in the former group. Although the mechanism is not entirely understood, these drugs are considered etiological factors for bruxism [35]. Disturbances in the central dopaminergic system have been linked to bruxism. Dopamine deficiency causes disinhibition of muscle activity, allowing motor restlessness and contractions of the jaws [36]. Indeed, bruxism induced by medications is considered a form of akathisia. While emotional stress is the leading cause of awake bruxism, sleep bruxism is centrally mediated [22,23]. As bruxism causes tooth wear and orofacial pain, and night guards frequently treat both symptoms, it is not surprising that we found a higher percentage of tooth wear and night guards in the study groups than in the controls. Our findings are in accordance with Yoshizawa et al. [37], who claim that tooth wear is a diagnostic tool for bruxism. However, other researchers disagree with this assumption [38].

We found that tooth wear was prevalent in both medicated and non-medicated Ci-PTSD groups, affecting more than 70% of patients. This percentage is much higher than the reported in the general population [39].

According to Schierz et al. [39], the overall prevalence of exposed dentin was 23.4% in the general population. However, it ranged from 12.9% in young women (20 to 29 years old) to 44.0% in older men (50 to 59 years old).

The present study results of tooth wear in 100/103 patients in PTSD patients are in accordance with our previous findings showing that 90.1% of Ci- PTSD patients presented tooth wear related to parafunctional habits [10].

The patients in the present study had poor oral hygiene and a very high caries level compared with the controls. Dental caries remains one of the most common chronic and multifactorial diseases worldwide. It results from a series of events that generally begins years before detecting the lesion. In addition, several health behaviors such as dietary, lifestyle, oral hygiene habits, and smoking also enter this casual chain [2,40,41].

PTSD patients have a significantly higher risk for nicotine dependency and marijuana or alcohol addiction than the general population [42,43]. Indeed, heavy smoking was more prevalent in our PTSD patients compared with the controls. [44].

PTSD patients included in our study had a high percentage of carious teeth. In the survey, the missing teeth component was 5.78 ± 6.95 in the study group versus 3.98 ± 4.89 in the control group indicates that Ci-PTSD patients were undertreated, although in Israel they have free and costless access to dental treatments. Indeed, in our study, the mean number of “filled” teeth (10.7) was higher, and the component of missing teeth (5.78) was lower than in the previously mentioned Israeli study. Still, our patients had a very high tendency to repeated carious activity. Possible etiological factors include chronic use of antidepressants, alcohol, and tobacco consumption and the resultant xerostomia [44,45,46]. A recent meta-analysis showed that SSRIs, SNRIs, and atypical antidepressants are all associated with an increased risk of dry mouth [47]. The poor oral hygiene, reflected in the high PI levels, also contributes to the high mean DMFt score of 19.97 in our Ci-PTSD patients. In their comprehensive review, Costa et al. found that the average DMFt score in the general population was 11.49 in 19–60-year-old patients [48], similar to our finding in the control group. More decayed and missing teeth were observed in children with combat-induced PTSD in Syria than in the general population [49]. In Israel, a high DMFt index of 23.8 was found in chronically institutionalized patients with psychiatric diseases [50].

Additionally, the number of dental implants was smaller in the study groups than in the control group.

This finding is surprising since, as mentioned above, combat-induced PTSD patients in Israel receive all dental treatments, including dental implants, free of charge. Most likely, dental anxiety, especially regarding surgical procedures and general neglect, leads to avoiding invasive surgical procedures among these patients [51]. Moreover, according to Lenk et al., PTSD is a significant predictor (OR = 16.836) of a high degree of dental fear [52]. Increased fear of dental treatment was ten times more prevalent in patients with PTSD [53].

### Limitations and Strengths

This study has several limitations. First, the sample size is relatively small. Yet, this is the most extensive study describing oral and facial manifestations in combat-induced PTSD patients to the best of our knowledge.

Additionally, as all the patients were referred for the treatment of active periodontal disease, the periodontal morbidity rate was 100% both in the PTSD and the control groups. A control group of combat-induced PTSD patients not referred for periodontal treatment might have been more appropriate for comparison.

A significant strength of the study is the demographic matching (age and gender) between the study and control groups which increases the power of the statistical analysis. Additionally, possible diagnostic confounders were controlled as the same clinician carried out all clinical examinations.

## 5. Conclusions

Within the limitations of this study, we can conclude:PTSD patients have high dental and oral pathologies, especially those treated with psychiatric medications.Our findings suggest that all patients with PTSD should undergo complete periodontal, oral, and dental examinations periodically and comprehensive education and training regarding proper oral hygiene.The use of night guards is frequently indicated in Ci-PTSD.The recognition of the particular needs of these patients concerning oral, periodontal, facial, and TMJ health can prevent dental complications and improve quality of life.

## Figures and Tables

**Figure 1 jcm-11-03249-f001:**
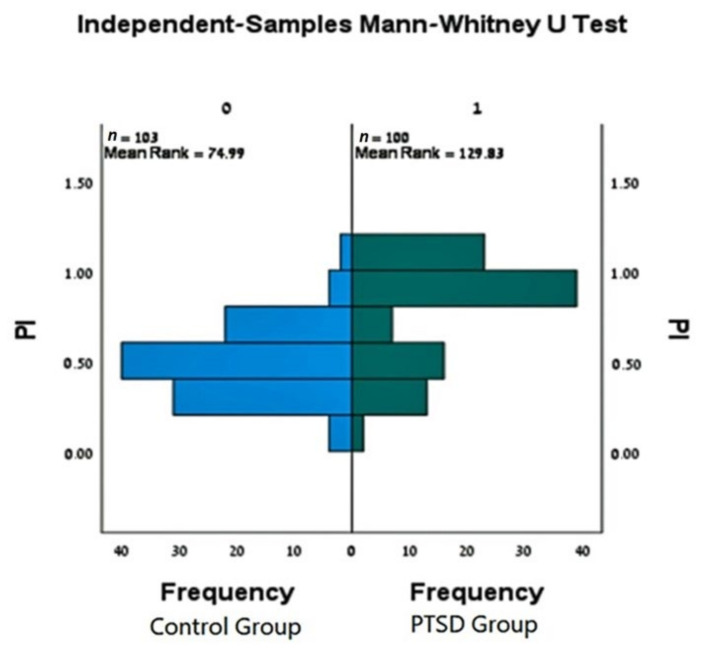
Plaque Index Frequency.

**Table 1 jcm-11-03249-t001:** A description and comparison of demographic and quantitative variables of PTSD patients with/without medications and a control group.

		PTSD	PTSD		Control	Significance	
		Total (*n* = 100)	With Medication (*n* = 40)	Without Medication (*n* = 60)	(*n* = 103)	Overall (ANOVA 3 Groups)	Post-Hoc (Tukey B Test)
Sex	Male	97 (97%)	39 (97.5%)	58 (97%)	102 (99%)		
	Female	3 (3%)	1 (2.5%)	2 (3%)	1 (1%)		
Age		60.55 ± 11.70	63.30 ± 10.79	56.43 ± 11.92	59.93 ± 13.29	*p* = 0.024	Med non- med *p* < 0.05
PI		0.72 ± 0.28	0.80 ± 0.24	0.67 ± 0.29	0.45 ± 0.29	*p* < 0.001 (*)	Any two groups of the 3 are different with *p* < 0.05
Decayed Missing Filled		3.47 ± 3.46	4.13 ± 4.63	3.03 ± 2.33	0.58 ± 1.24	*p* < 0.001	- “ -
		5.78 ± 6.95	5.60 ± 7.80	5.90 ± 6.38	3.98 ± 4.89	*p* = 0.102	N/S
		10.70 ± 5.63	9.65 ± 6.60	11.4 ± 4.85	8.50± 4.76	*p* = 0.003	Control–non- med *p* < 0.05
DMF		19.97 ± 8.07	19.48 ± 9.28	20.30 ± 7.22	13.05 ± 6.23	*p* < 0.001	Control–PTSD *p* < 0.05
Implants		1.15 ± 2.12	1.08 ± 2.011	1.25 ± 2.30	2.0 ± 3.21	*p* = 0.085	N/S

(*) The significance of *p* < 0.001 for PI 3 groups differences was found in the ANOVA Test and the Kruskal-Wallis Non-Parametric Test (performed due to PI’s non-Normal distribution).

**Table 2 jcm-11-03249-t002:** Examined Variables—Counts and Tests Significance of Differences and Associations.

		PTSD		Control	Test #Conditions × #Groups Significance
		With Medication	Without Medication		
Smoking status	Never	16 (40.0%)	43 (71.7%)	71 (68.9%)	Pearson Chi-Square 4 × 3 *p* < 0.001
	Heavy	22(55.0%)	13 (21.7%)	16 (15.5%)	
	Light	0 (0.0%)	1 (1.7%)	10 (9.7%)	
	Former	2 (5.0%)	3 (5%)	6 (5.8%)	
Periodontal disease status: Severity	Mild- Moderate (Stages I or II)	15 (37.5%)	42 (70.0%)	75 (72.8%)	Pearson Chi-Square 2 × 3 *p* < 0.001
	Severe (Stages III or IV)	25 (62.5%)	18 (30.0%)	28 (27.2%)	
Grading	B	12 (30.0%)	25 (41.7%)	35 (34.0%)	Non-Significant
	C	22 (55%)	21 (35.0%)	28 (27.2%)	Non-Significant
Local– General	Localized	12 (30%)	26 (43.3%)	53 (51.5%)	Pearson Chi-Square 2 × 3 *p* = 0.066
	Generalized	28 (70%)	34 (56.7%)	50 (48.5)	
Tooth wear (edentulous patients not included)	None	2 (5.6%)	3 (5.1%)	27 (26.5%)	Pearson Chi-Square 3 × 3 *p* < 0.001
	Mild	8 (22.2%)	13 (22%)	42 (41.2%)	
	Severe	26 (72.2%)	43 (72.9%)	33 (32.4%)	
Night guard (no edentulous patients)	No	12 (33.3%)	34 (57.6%)	94 (92.2%)	Pearson Chi-Square 2 × 3 *p* < 0.001
	Yes	24 (66.7%)	25 (42.4%)	8 (7.8%)	
TMJ/Facial pain	No	12 (30.0%)	31 (51.7%)	90 (88.2%)	Pearson Chi-Square 2 × 3 *p* < 0.001
	Yes	28 (70.0%)	29 (48.3%)	12 (11.8%)	

## Data Availability

The corresponding author has all the data.

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
