# Peer review of "Oral and Dental Considerations of Combat-Induced Post Traumatic Stress Disorder (PTSD)—A Cross-Sectional Study"

_jcm, 2022, doi:10.3390/jcm11113249_

Round 1

Reviewer 1 Report

jcm-1710122

Oral and Dental Considerations of Combat-induced PTSD - a Comparative Study

100 Israeli veterans diagnosed with combat-induced post-traumatic stress disorder (Ci-PTSD) and 103 age and sex matched non-Ci-PTSD periodontal patients were examined for periodontitis, caries, tooth wear and temporomandibular problems. 40 Ci-PTSD patients were under psychiatric medication. Oral morbidity (caries, periodontal disease, tooth wear) was more frequent or severe in Ci-PTSD patients than non Ci-PTSD controls. Ci-PTSD patients under psychiatric medication exhibited more morbidity than non-medicated.

This is an interesting and relevant study on the association of Ci-PTSD and oral diseases. However, some issues are unclear and have to be clarified.

Comments:

Title

Replace “a comparative” by “a cross-sectional”.

Affiliations

What does Tel Aviv University mean? What department or institute of Tel Aviv University? The authors may wish to be more precise.

Abstract

Page 1, lines 19, 21: Please provide p values.

Material & Methods

Page 2, lines 67-69: Please provide reference number of ethics approval.

Page 2, line 81: Replace “Intra” by “intra”.

Page 2, lines 84-94: What about bleeding on probing (BOP)?

Pages 2-3, lines 77-104: Extraoral examination should be performed prior to intraoral, should it not?

Page 3, line 106-112: Periodontal diagnosis encompasses stage and grade. The authors may wish to provide grade.

Results

Page 4, lines 140-148: If variables are not normally distributed no parametric tests must be used. What test did the authors use to reveal distribution of data? Are the other variables normally distributed?

Page 5, Table 2, lines 170-173: What does mild and severe periodontitis mean? The 2018 classification distinguishes between mild (stage I), moderate (stage II) and severe (stages III and IV) periodontitis. I strongly encourage the authors to use stages to characterize periodontitis. If they want to limit comparison to 2 groups, they may fuse stage I and II as well as stage III and IV. However, mild and severe is inappropriate.

Page 5, lines 174-176: The difference is not significant. Delete passage.

Page 6, lines 184-195: This is all provided in Tale 2 already. Delete passage.

Discussion

Page 6, lines 204: Zhan et al. 2014 present 9 different prevalence of severe periodontal disease in Germany. 20% is only one rate of 9 different rates. 20% is based on SHIP-0. DMS V provides actual numbers and not DMS IV that is used by Zhan et al. 2014.

Page 6, lines 216-219: Traumatic occlusal forces may be a local factor aggravating biofilm induced periodontal disease. The effect on gingival recession is at least controversial.

Page 7, lines 252-267: This passage should follow the passage starting with “The patients in the present study…”. Passage on nicotine dependency should follow later.

Tables

Page 5: The authors may add p values for comparisons between single groups (Ci-PTSD with and without medication, control).

Author Response

May 25, 2022

Dear Prof. Chaushu

Editor of the special issue New Frontiers and Boundaries in the Use of Biomaterials in Dentistry

Thank you for the review of our manuscript entitled Oral and Dental Considerations of Combat-induced PTSD - a Comparative Study- jcm-1710122

Comments from the Reviewers:

100 Israeli veterans diagnosed with combat-induced post-traumatic stress disorder (Ci-PTSD) and 103 age and sex matched non-Ci-PTSD periodontal patients were examined for periodontitis, caries, tooth wear and temporomandibular problems. 40 Ci-PTSD patients were under psychiatric medication. Oral morbidity (caries, periodontal disease, tooth wear) was more frequent or severe in Ci-PTSD patients than non Ci-PTSD controls. Ci-PTSD patients under psychiatric medication exhibited more morbidity than non-medicated.

This is an interesting and relevant study on the association of Ci-PTSD and oral diseases. However, some issues are unclear and have to be clarified.

Comments: reviewer 1

Title

Repliace “a comparative” by “a cross-sectional”.

Correct- The title has been changed

Affiliations

What does Tel Aviv University mean? What department or institute of Tel Aviv University? The authors may wish to be more precise.

Thank you- we wrote again in detail all the affiliations

Abstract

Page 1, lines 19, 21: Please provide p values.

We have added the p-values

Material & Methods

Page 2, lines 67-69: Please provide reference number of ethics approval.

The ethical approval has no reference number; however, we have attached the IRB approval. It was issued on August 15, 2019.

Page 2, line 81: Replace “Intra” by “intra”.

Replaced

Page 2, lines 84-94: What about bleeding on probing (BOP)?

We did calculate the BOP, however, due to the high rate of smokers (especially in the test group), BOP results were considered misleading, accordingly, they were not included in the report.

Pages 2-3, lines 77-104: Extraoral examination should be performed prior to intraoral, should it not?

Of course, we have changed the order in the manuscript as well.

Page 3, line 106-112: Periodontal diagnosis encompasses stage and grade. The authors may wish to provide grade.

A complete periodontal diagnosis was performed according to the staging and grading, however, the outcome was a large amount of groups that did not allow for a valuable statistical analysis. Therefore, it was decided not to include the grading, whichwe consider is less relevant for a cross-section study.

Results

Page 4, lines 140-148: If variables are not normally distributed no parametric tests must be used. What test did the authors use to reveal distribution of data? Are the other variables normally distributed?

Fig. 1 shows the Mann-Whitney U test performed for the abnormally distributed values of the PI. (Explained in lines 171-174). All other parameters were normally distributed.

Page 5, Table 2, lines 170-173: What does mild and severe periodontitis mean? The 2018 classification distinguishes between mild (stage I), moderate (stage II) and severe (stages III and IV) periodontitis. I strongly encourage the authors to use stages to characterize periodontitis. If they want to limit comparison to 2 groups, they may fuse stage I and II as well as stage III and IV. However, mild and severe is inappropriate.

We have corrected and explained (as we did in our data collection), that mild periodontitis is stage 1 and 2 and severe periodontitis is stage 3 and 4.

Page 5, lines 174-176: The difference is not significant. Delete passage.

Deleted

Page 6, lines 184-195: This is all provided in Tale 2 already. Delete passage.

 Deleted

Discussion

Page 6, lines 204: Zhan et al. 2014 present 9 different prevalence of severe periodontal disease in Germany. 20% is only one rate of 9 different rates. 20% is based on SHIP-0. DMS V provides actual numbers and not DMS IV that is used by Zhan et al. 2014.

We have explained and gave the different rates according to Zhan 2014. Unfortunately, the DSM V could not be found.

Page 6, lines 216-219: Traumatic occlusal forces may be a local factor aggravating biofilm induced periodontal disease. The effect on gingival recession is at least controversial.

Deleted

Page 7, lines 252-267: This passage should follow the passage starting with “The patients in the present study…”. Passage on nicotine dependency should follow later.

Corrected

Tables

Page 5: The authors may add p values for comparisons between single groups (Ci-PTSD with and without medication, control).

The P values are in the table (marked).

Reviewer 2 Report

The article entitled “Oral and Dental Considerations of Combat-induced Post Traumatic Stress Disorder (PTSD) - a Comparative Study” aimed to compare dental, periodontal, oral, and joint/muscle tenderness among Israeli combat-induced post-traumatic stress disorder (Ci-PTSD) war veterans to non-PTSD patients The paper is in line with journal’s aim, moreover, Authors have well revised several issues; however, I ask authors to add aimed to provide an overview of available silver-treated dental implants and  some key concepts.

-       The introduction section is too short, the authors must introduce the issue of periodontitis under various aspects, from the difficulty in the diagnosis, treatment and prognosis of the dental elements, in fact, non-surgical therapy is sometimes not enough and surgical therapy is used when the pocket has a depth greater than 5mm and is bleeding (please see and discuss DOI: 10.1002 / JPER.20-0305) and to perform surgery it is necessary that the systemic conditions of the candidate patient are optimal, hence the problem of the link between periodontitis and the stressful conditions that induce an increase in cortisol with worsening and acceleration of bone loss, it is therefore necessary to better discuss the choice of the topic addressed.

-       Why didn't the authors use the new classification of periodontal disease?

-       Conclusions cannot be reduced to a sentence: you must improve them highlighting the limits and the future insights pointed out from this article.

-       The formatting of the references is not correct, please check the journal instructions for authors

-       Several moderate typos are present in the text, please, amend

Author Response

May 25, 2022

Dear Prof. Chaushu

Editor of the special issue New Frontiers and Boundaries in the Use of Biomaterials in Dentistry

Thank you for the review of our manuscript entitled Oral and Dental Considerations of Combat-induced PTSD - a Comparative Study- jcm-1710122

Comments from the Reviewers:

 The article entitled “Oral and Dental Considerations of Combat-induced Post Traumatic Stress Disorder (PTSD) - a Comparative Study” aimed to compare dental, periodontal, oral, and joint/muscle tenderness among Israeli combat-induced post-traumatic stress disorder (Ci-PTSD) war veterans to non-PTSD patients The paper is in line with journal’s aim, moreover, Authors have well revised several issues; however, I ask authors to add aimed to provide an overview of available silver-treated dental implants and  some key concepts.

 -       The introduction section is too short, the authors must introduce the issue of periodontitis under various aspects, from the difficulty in the diagnosis, treatment and prognosis of the dental elements, in fact, non-surgical therapy is sometimes not enough and surgical therapy is used when the pocket has a depth greater than 5mm and is bleeding (please see and discuss DOI: 10.1002 / JPER.20-0305) and to perform surgery it is necessary that the systemic conditions of the candidate patient are optimal, hence the problem of the link between periodontitis and the stressful conditions that induce an increase in cortisol with worsening and acceleration of bone loss, it is therefore necessary to better discuss the choice of the topic addressed.

We have added a paragraph explaining the periodontal disease, diagnosis, and treatment.

-       Why didn't the authors use the new classification of periodontal disease?

We did use the new classification, and clarified its use.

-       Conclusions cannot be reduced to a sentence: you must improve them highlighting the limits and the future insights pointed out from this article.

We have expanded the strength and limitation paragraph.

-       The formatting of the references is not correct, please check the journal instructions for authors

All the reference section was corrected according to the instructions.

-       Several moderate typos are present in the text, please, amend

Corrected

Round 2

Reviewer 1 Report

jcm-1710122R1

Oral and Dental Considerations of Combat-induced PTSD - a Comparative Study

100 Israeli veterans diagnosed with combat-induced post-traumatic stress disorder (Ci-PTSD) and 103 age and sex matched non-Ci-PTSD periodontal patients were examined for periodontitis, caries, tooth wear and temporomandibular problems. 40 Ci-PTSD patients were under psychiatric medication. Oral morbidity (caries, periodontal disease, tooth wear) was more frequent or severe in Ci-PTSD patients than non Ci-PTSD controls. Ci-PTSD patients under psychiatric medication exhibited more morbidity than non-medicated.

This is the revision of manuscript jcm-1710122 reporting an interesting and relevant study on the association of Ci-PTSD and oral diseases. The authors have addressed most comments However, some issues are still unclear and have to be clarified.

Comments:

Material & Methods

Page 3, line 135-142: Periodontal diagnosis encompasses stage and grade. The authors may wish to provide grade. Grade will not split the sample into too small groups. To describe groups properly with respect to periodontal conditions, the authors should provide frequency of grade B and C in Ci-PTPS and control (Table 2).

Results

Page 5, Table 2: The 2018 classification distinguishes between mild (stage I), moderate (stage II) and severe (stages III and IV) periodontitis. Thus, by definition stage II is not mild but moderate periodontitis. I strongly encourage the authors to use stages to characterize periodontitis. If they want to limit comparison to 2 groups, they may fuse stage I and II as well as stage III and IV. However, whereas severe is appropriate for stage III and IV, mild is appropriate only for stage I not II.

Author Response

Dear reviewer,

Thank you for your help in ameliorating our manuscript.

We have added according to your recommendations:

  1. We changed the definition of the severity of the periodontal disease.
  2. We added to table 2 the results of the grades B and C. 
  3. We hope that the change fulfills your comments.
  4.